# Spatio-Temporal Evolution and Influence Mechanism of Habitat Quality in Guilin City, China

**DOI:** 10.3390/ijerph20010748

**Published:** 2022-12-31

**Authors:** Yunlin He, Yanhua Mo, Jiangming Ma

**Affiliations:** 1Key Laboratory of Ecology of Rare and Endangered Species and Environmental Protection, Ministry of Education, Guangxi Normal University, Guilin 541006, China; 2Guangxi Key Laboratory of Landscape Resources Conservation and Sustainable Utilization in Lijiang River Basin, Guilin 541006, China; 3Institute for Sustainable Development and Innovation, Guangxi Normal University, Guilin 541006, China

**Keywords:** habitat quality, landuse change, InVEST model, geographical detector, Guilin City

## Abstract

Based on the models of ArcGIS10.5, Fragstats 4.2, and InVEST, this research describes the temporal and spatial evolution characteristics of habitat quality in Guilin from three aspects, which are land use change, landscape pattern change, and habitat quality evaluation, and further explores the main driving factors of Guilin’s habitat quality change by using the method of geographic detector evaluation. The results indicate that from 2000 to 2020, the land use type in Guilin City is dominated by forest, accounting for the highest proportion of 77.87%. The forest has decreased significantly, the mutual transformation of forest and cropland is obvious, and the area of impervious has continued to increase. A large amount of cropland is occupied, indicating that human activities were the main factor in land use transformation. From 2000 to 2020, the irregularity of the patch shape of each land use type was deepened, the fragmentation degree was relatively stable, the landscape diversity was enhanced, and the spatial distribution of each patch showed a relatively obvious heterogeneity. From 2000 to 2020, the habitat quality of Guilin City was mainly high-grade and the habitat quality was good, but the overall trend showed a downward trend, and the spatial difference was obvious. From 2000 to 2020, elevation, normalized difference vegetation index (NDVI), splitting index (SPLIT), and slope were the main factors affecting the habitat quality of Guilin City, among which elevation and NDVI had the most significant effects.

## 1. Introduction

The assessment and monitoring of habitat quality (HQ) is an important part of the 15th United Nations Sustainable Development Goal (SDG15), as well as an important indicator of the construction and performance evaluation of the innovation demonstration area of the national sustainable development agenda. Habitat quality is usually defined as the ability of the environment to provide suitability and sustainable living conditions for individual life and population. It is an important ecosystem service [1] and an important embodiment of biodiversity [2]. SDG15 pointed out that we should curb the loss of biodiversity and incorporate ecosystem and biodiversity values into national and local planning. The transformation process of land use types is a response process of land use patterns to changes in social and economic development stages [3]. Relevant research shows that [4,5], with the advance of urbanization and the increase in population, the impact of human beings on the natural ecological environment is growing [6], resulting in the decline of forest coverage [7], the destruction of the ecological environment, and the degradation of habitats to varying degrees [8]. The impact of human activities on the ecological environment is mainly reflected in the change in land use mode [9]. The research on the Spatio-temporal pattern evolution of habitat quality under the impact of land use change in local cities can quantitatively display the status of habitat quality, promote the protection and restoration of ecological environments in ecologically vulnerable areas, as well as optimize regional ecological layouts.

At present, habitat quality assessment methods can be mainly divided into two categories: one is based on field measurement data modeling assessment [10,11,12], and the other is based on model landscape pattern assessment [13,14,15]. The measured data have the characteristics of high precision and strong pertinence, which can more accurately reflect the local habitat quality status, but its limitations are also very obvious. As for the current situation, the field observation can only be carried out on a small scale due to the limitation of the number of samples and data. The model-based regional landscape pattern research fully integrates various characteristics of the ecosystem. Compared with the disadvantages of time-consuming and expensive field measurement methods, model-based methods can complete a wide range of research and have the advantages of a short cycle, a large amount of information, sustainability, and high-cost performance [16]. Among many habitat quality research models, such as the Habitat Suitability Model (HSI) [17], Maximum Entropy Model (MxEnt) [18], and Artificial Intelligence Assessment Model for Ecosystem Service Functions (ARIES), Integrated Valuation of Environmental Services and Trade-offs (InVEST model) jointly developed by Stanford University, Nature Conservancy (TNC) and the World Wildlife Fund (WWF) is the most mature and widely used [17,19]. Currently, the time and spatial scale of research on the quality of Guilin’s habitat are relatively short [20,21]. Large-scale research is mostly based on multiple indicators or ecosystem services to evaluate the quality of Guilin’s habitat [22,23,24]. Research on long-term time series using models is still lacking, especially research on its impact mechanism. The model-based habitat quality research method integrates remote sensing and geographic information technology. Although its accuracy in a small range is lower than that of the evaluation method based on measured data, it can reveal the macro characteristics of habitat evolution from a large scale, has a good spatial visualization effect, and can better analyze the influencing factors in a large range.

Rocky desertification is a typical feature of ecologically fragile areas, while Guilin, as a typical karst landform distribution area and world-class tourist city, is an important ecological security barrier to the fragile karst environment in southern China [25]. Based on land use, this research calculates the habitat quality of Guilin, analyzes the distribution and transformation of different levels of habitat quality, selects indicators from both natural and socio-economic aspects, and uses geographic detectors to detect the driving factors that affect the spatial differentiation and temporal evolution of habitat quality. This study aims to serve as a reference for formulating policies about land management and ecological protection in Guilin, as well as providing a scientific basis for improving the quality of regional habitats, restoring karst areas ecologically, conserving landscape resources, and developing sustainable communities.

## 2. Materials and Methods

### 2.1. Study Area

Guilin is located in the northeast of Guangxi Zhuang Autonomous Region, in the center of four provinces and regions of Guangxi, Hunan, and Guizhou, with an administrative area of 2.78 × 104 km^2^, longitude and latitude range is (109°36′50″ 111°29′30″ E, 24°15′23″ 26°23′30″ N) [22], belonging to the subtropical monsoon climate zone, with mild and rainy climate and rich tourism resources (Figure 1). Guilin is an important node connecting South China and Central China, West China and East China, and inland and coastal areas. It is also an international tourist city. Has unique Lijiang River landscape is a world-renowned scenic spot on the southwest border of China [23].

### 2.2. Data Resources

Using the land use data from the research titled “30 m annual land cover and its dynamics in China from 1990 to 2019”, jointly written by Professor Yang Jie and Huang Xin of Wuhan University [26], which has a resolution of 30 m, this study has been conducted. The land use data of Guilin City include cultivated land, forest land, shrubs, grassland, water area, wasteland, and construction land. The 30 m resolution elevation data come from the Geospatial Data Cloud (http://www.gscloud.cn/, accessed on 30 September 2022). DEM was used to calculate the slope and aspect data; natural environment data come from the China Meteorological Administration, Chinese Academy of Sciences (http://www.resdc.cn/, accessed on 12 October 2022), and National Earth System Science Data Center (http://www.geodata.cn/, accessed on 15 October 2022). Social and economic data are from the Statistical Yearbook, the National Earth System Science Data Center, and the Resource and Environmental Science Data Center of the Chinese Academy of Sciences. The Splitting index is calculated from multi-year land use data.

### 2.3. Methods

#### 2.3.1. Landscape Pattern Index

The landscape pattern index can highly concentrate the landscape pattern information. It is a simple quantitative index reflecting the internal structure composition and spatial configuration characteristics of the landscape and is a common method to study the changes and laws of landscape patterns [27,28]. In order to explore the relationship between various landscape types and the degree of human interference, this study used Fragstats 4.2 software (Landscape Ecology Lab, University of Massachusetts - Amherst, Amherst, America) to select nine representative landscape pattern indexes from two aspects of landscape structure characteristics and landscape space characteristics, including landscape shape index, edge density, patch density, mean patch fractal dimension, splitting index, Shannon’s evenness index, Shannon’s diversity index, contagion index data, and aggregation index [29,30,31,32]. Quantitative analysis of landscape structure composition and its spatial configuration characteristics. The landscape indexes and their ecological meanings are shown in (Table 1).

#### 2.3.2. InVEST Model

InVEST (Integrated Valuation of Environmental Services and Trade-offs) model is an open-source ecosystem service function assessment model jointly developed by Stanford University, the Nature Conservancy (TNC), and the World Wide Fund for Nature (WWF) [33,34]. Habitat quality directly reflects the quality of biological habitats, indicating the ability of ecosystems to provide suitable production conditions for the survival of individual organisms [35]. In this study, InVEST 3.12.0 software was used to analyze Guilin’s land use cover and habitat threat factor information from 2000 to 2020. A habitat quality map was formed to simulate and evaluate the habitat quality. Habitat degradation degree refers to the degradation degree of habitat after being affected by threat factors. The effects of threat factors on a habitat can be divided into linear and exponential types, and the calculation formula is as follows [34,35,36]:(1)$$i_{rxy}=\{\begin{array}{l} 1-\left(\frac{d_{xy}}{d_{rmax}}\right)\;\mathrm{if}\;\mathrm{linear}\;\\ exp\left[-\left(\frac{d_{xy}}{d_{r\;max}}\right)\times d_{xy}\right]\;\mathrm{if}\;\mathrm{exponential}\; \end{array}$$

*i_rxy_* is the influence of threat factor *r* in grid *y* on grid *x*; *d_xy_* is the linear distance between grid *x* and *y*; *d_r max_* is the maximum range of threat *r*. By referring to the relevant literature and relevant research results [33,34,35,36,37,38], the values are assigned according to the situation of Guilin City and experts’ suggestions. Cultivated land, wasteland, and construction land are selected as threat factors. The weight of threat factors and their impact distance are shown in (Table 2).

According to the relative sensitivity of each habitat type to each threat factor, the habitat degradation degree of grid *x* in land use type *j* is shown in Table *D_xj_*, and the formula is as follows:(2)$$D_{xj}=\overset{R}{\underset{r=1}{\sum }}\overset{Y_{r}}{\underset{y=1}{\sum }}\frac{W_{r}}{\sum_{r=1}^{R}W_{r}}r_{y}i_{rxy}\beta_{x}S_{jr}$$

*D_xj_* is the total threat level of grid *x* in land use type *j*; *R* is the habitat threat factor; *r* is all grids of *r* threat source; *Y_r_* refers to a group of grids threatened by *r*; *W_r_* is the weight of different threat factors *r*; *r_y_* is the threat intensity of grid *y*; *i_rxy_* is the threat level of *ry* to habitat grid *x*; *β_x_* is the accessibility level of grid *x*; *S_jr_* is the sensitivity of habitat *j* to threat factor *r*. The habitat degradation degree is between 0 and 1, and the larger the value is, the more obvious the habitat degradation degree is.

Habitat quality value is a dimensionless value for evaluating regional habitat quality. In land use type *j*, the habitat quality of grid *x* is expressed by *Q_xj_*, and the formula is as follows:(3)$$Q_{xj}=H_{j}\left(1-\left(\frac{D_{xj}^{z}}{D_{xj}^{z}+k^{z}}\right)\right)$$

According to the use description of InVEST model, the actual situation of the study area and relevant research results [34,35,36,37,38] determine the habitat suitability of different land use types and the relative sensitivity of each habitat type to each threat factor, as shown in (Table 3).

*Q_xj_* is the habitat quality index; *H_j_* is the habitat suitability of habitat type *j*; *K* is the semi-saturation constant; *Z* is a normalized constant, and the habitat quality value is 0–1. The higher the value, the better the habitat quality.

#### 2.3.3. Geographical Detector

Geographic detectors are a set of statistical methods to detect spatial heterogeneity and reveal its driving force. They were proposed by Wang Jinfeng et al. [39] and are often used to study the influencing factors and mechanisms of spatial stratification heterogeneity. Its core idea is based on the following assumption: if an independent variable has an important influence on a dependent variable, the spatial distribution of the independent variable and the dependent variable should be similar. The geographic detector mainly includes four modules: ecological detector, interaction detector, risk detector, and differentiation and factor detector [40,41,42,43,44]. This study mainly uses its factor detector to detect the spatial stratification heterogeneity of habitat quality and the degree of interpretation of different factors on the spatial differentiation of habitat quality. Its explanatory power is expressed by the *q* value. According to the *q* value, we can see the impact of each factor on the habitat quality to know the dominant factors affecting the habitat quality in Guilin. The expression of the *q* value is [45,46,47,48]:(4)$$q=1-\frac{\sum_{h=1}^{L}N_{h}\sigma_{h}^{2}}{N\sigma^{2}}=1-\frac{W_{SS}}{T_{SS}}$$
(5)$$W_{SS}=\overset{L}{\underset{h=1}{\sum }}N_{h}\sigma_{h}^{2}$$
(6)$$T_{SS}=N\sigma^{2}$$

*q* is the explanatory power of each influencing factor to the habitat quality. The value is 0–1. The higher the value is, the stronger the explanatory power of the factor is; *h* = 1, 2, …; *L* represents the stratification of independent variable *Y* or factor *X*; *N_h_* and *N* are the unit numbers of layer *h* and the whole area, respectively; *σ*^2^ is the variance of the population; *σ^2^* is the variance of layer *h*; and *W_SS_* and *T_SS_* are the sums of the variance within the layer and the total regional variance, respectively.

Concerning the relevant literature [30,36,38] and the actual situation of the study area, NDVI, elevation, slope, aspect, mean annual precipitation (MAP), mean annual air temperature (MAAT), gross domestic product (GDP), population (POP), night light (NL), and SPLIT are selected as independent variables in this study, and habitat quality value is taken as the dependent variable. A grid with a side length of 5 km is established in the study area, and the grid center point is taken as the sampling point. The impact factors and habitat quality are reclassified using the natural breakpoint method, It is divided into nine categories, and the driving factors of habitat quality in 2000, 2005, 2010, 2015, and 2020 are analyzed, respectively.

## 3. Result and Analysis

### 3.1. Temporal and Spatial Changes in Land Use

The distribution and change in land use in Guilin from 2000 to 2020 are shown in (Figure 2 and Figure 3). The land use change in Guilin is analyzed by combining the land use distribution change map (Figure 2) and the land use change flow chart (Figure 3) in each period. It can be seen from (Figure 2) that forest is widely distributed in the whole area of Guilin City and is the main type of land use; cropland is distributed along the contour line in areas with low altitude, and its distribution range is only second to that of the forest. Other land use types are scattered and small in area. In Figure 3, the mutual conversion between cropland and forest can be seen, while other land types are less obvious.

The transfer matrix is used to further explore the changes among different land use types in Guilin. As shown in (Table 4), forest land is the type of land use with the greatest degree of change in Guilin from 2000 to 2020. The forest land is largely converted into cropland, and 1373.463 km^2^ of the forest is converted into cropland. It can be seen that human activities to destroy forests and open up wasteland are the main factors for the increase in impervious areas. At the same time, there are also large areas of cropland converted to forest and impervious, which are 777.5 km^2^ and 156.075 km^2^, respectively, indicating that human activities have a significant impact on the transformation of land use types. As the proportion of the barren area is too small, it is not particularly obvious in the transfer matrix.

As a typical landscape resource-based city, Guilin is famous for its mountains and waters. It can also be seen from (Table 5) that the most important type of land use in Guilin is forest. The proportion of forest area in the study period is 76.79% at the lowest and 78.87% at the highest. From 2000 to 2020, the change in forest land is also the most obvious, with the area proportion decreasing by 1.98%, which is much higher than other types of land. The second is cropland. The area proportion increased significantly during the study period, with the highest proportion reaching 20.88%. From 2000 to 2020, the area proportion increased by 1.75%. However, it can be seen from the table that the proportion of impervious decreased significantly from 2015 to 2020, and the proportion of forests increased, indicating that Guilin’s ecological protection policies, such as returning farmland to forests, achieved some results from 2015 to 2020.

### 3.2. Landscape Pattern Change

The landscape metrics module in Fragstats 4.2 is used to select nine indicators for landscape pattern analysis, namely, landscape shape index (LSI), edge density (ED), patch density (PD), mean patch fractal dimension (FRAC_MN), splitting index (SPLIT), Shannon’s diversity index (SHDI), Shannon’s evenness index (SHEI), contagion index (CONTAG), and aggregation index (AI). The results are shown in (Figure 4) and (Table 6). From 2000 to 2020, PD and LSI decreased first and then increased, indicating that the degree of landscape fragmentation in Guilin decreased first and then increased. LSI changed from 101.9 in 2000 to 115.5 in 2020, and the degree of patch irregularity increased significantly. The ED value is increasing, mainly due to the fragmentation of landscape patches caused by human activities and the increase in boundary density. The values of SHDI and SHEI are increasing, indicating that the land use richness has increased and the diversity of the landscape has been enhanced during the study period. The SHEI value is lower, indicating that the landscape pattern uniformity is poor. The AI and CONTAG values are continuously decreasing, indicating that the overall landscape separation degree of Guilin City is increasing, and the aggregation degree is weakening. In general, from 2000 to 2020, the irregularity of the shape of each land use patch was deepened, the fragmentation degree was relatively stable, the landscape diversity was enhanced, and the spatial distribution of each patch showed obvious heterogeneity.

### 3.3. Assessment of Habitat Quality in Guilin City

In this study, InVEST model is used to calculate the habitat quality of Guilin. To better analyze the spatial and temporal distribution characteristics of the habitat quality in Guilin, this study divides the calculated results of the habitat quality into five categories at equal intervals: low (0–0.2), lower (0.2–0.4), medium (0.4–0.6), higher (0.6–0.8), and high (0.8–1). The spatial and temporal distribution of habitat quality in Guilin from 2000 to 2020 is shown in (Figure 5). It can be seen from the figure that the overall habitat quality in Guilin has declined significantly. It is shown that the low-grade quality habitats are expanding and the area is increasing, while the high-grade quality habitats are shrinking and the area is decreasing. As shown in (Figure 6), the proportion of high-quality habitats decreased from 54.5% to 49.4%, and that of low-quality habitats increased from 10% to 13.5%. The proportion of higher and medium-quality habitats was relatively small. In general, the habitat quality of Guilin is mainly of high grade, and the habitat quality is good.

The spatial differentiation of habitat quality is obvious in Guilin. The habitat quality in the middle, northeast, and southwest is poor, especially in Guilin urban area. The habitat quality in the northwest is good, mainly high-quality habitat. The spatial distribution map of habitat quality changes in Guilin from 2000 to 2020 (Figure 7) is obtained by superimposing and analyzing the habitat quality distribution maps in different years. It can be seen from the map that the areas with decreased habitat quality from 2000 to 2005 are mainly distributed in the middle and southwest. From 2005 to 2010, the habitat quality of some regions in the southwest and northwest of Guilin was improved, and the degraded areas were scattered. From 2010 to 2015, habitat degradation intensified, concentrated in the middle and southwest. From 2015 to 2020, the degradation trend will slow down, concentrated in the central and southwest. In general, from 2000 to 2020, the area of habitat degradation in Guilin will be larger than the area of improvement, and the degradation areas will be concentrated in the middle and south.

### 3.4. Driving Forces Analysis

In order to further explore the changes in the habitat quality in Guilin, this study uses the geographical detector analysis method and selects the NDVI, elevation, slope, aspect, MAAT, MAT, GDP, POP, NL, and SPLIT of Guilin from 2000 to 2020 as independent variables, and the habitat quality value as the dependent variable. Finally, the explanatory power q of each driving factor in each year is obtained. It can be seen from (Figure 8) that the interpretation degree of each influencing factor to the habitat quality in 2000 is different, in which elevation > NDVI > SPLIT > NL > slope > GDP > POP > MAAT > MAP > aspect, it can be seen that elevation was the main driving factor of the habitat quality in 2000. In 2005, elevation > NDVI > SPLIT > slope > NL > POP > GDP > MAAT > MAP > aspect was the dominant factor in the interpretation degree of each influencing factor to the habitat quality. Elevation > NDVI > SPLIT > NL > slope > GDP > POP > MAAT > MAP > aspect is also the dominant factor in the interpretation of influencing factors of habitat quality in 2010. In 2015, elevation > NDVI > SPLIT > slope > GDP > MAAT > NL > population > MAP > aspect was the dominant factor in the interpretation of factors affecting habitat quality. At the same time, the explanatory power of NL decreased. NDVI > elevation > SPLIT > slope > NL > GDP > POP > MAAT > MAP > aspect is the most powerful factor in the interpretation of influencing factors of habitat quality in 2020.

From 2000 to 2020, the explanatory power of each habitat quality influencing factor has changed to a certain extent, and elevation has always been an important influencing factor of Guilin’s habitat quality. Among the natural factors, the explanatory power of elevation, NDVI, and slope is higher, while the explanatory power of aspect, MAP, and MAAT is lower. Among social factors, SPLIT and NL have relatively high explanatory power, while POP and GDP have relatively low explanatory power. SPLIT has a high explanatory power and shows an upward trend, which indicates that the intensification of landscape pattern fragmentation has damaged the ecological environment to a certain extent and inevitably affected the regional habitat quality. Because of the undulating terrain of Guilin, the forest land is mostly distributed in the areas with high altitudes, and the impervious is distributed in the areas with low altitudes along the contour line, so the elevation is the main factor affecting the habitat quality of Guilin. In general, the factors affecting Guilin’s habitat quality are mainly natural and relatively stable. The habitat quality is closely related to elevation, NDVI, SPLIT, slope, and NL.

## 4. Discussion

The transformation process of land use type is the response process of land use form to the change in the social and economic development stage [27]. From 2000 to 2020, forest land has always been the most important type of land use in Guilin, which fully reflects the characteristics of Guilin as a typical landscape resource-based city and a world-class tourist city. However, with the development of urbanization, the area of cultivated land and construction land has continued to grow, while the area of shrubs, grasslands, and waters has declined. Therefore, it is necessary to further analyze the driving factors of land use change based on the research results. With the development of the economy and society, human activities have had an indelible impact on the types of land use and landscape patterns in Guilin. From 2000 to 2020, the irregular shape of each land use patch was deepened, the landscape diversity was enhanced, and the spatial distribution of each patch showed obvious heterogeneity. The area and shape of patches will affect regional biodiversity and ecosystem service supply, and habitat quality is the support condition of biodiversity and the basis of regional ecological security. The average proportion of high-grade habitat quality in Guilin is 52%, and the overall habitat quality is good. From 2000 to 2020, the proportion of high-grade quality habitats decreased from 55% to 49%, and the proportion of low-grade quality habitats increased from 10% to 13%. The overall habitat quality showed a downward trend. The rate of habitat decline varies in different years. The most obvious period of habitat quality decline is from 2010 to 2015, which is also a period of rapid economic and urbanization development in Guilin. The continuous expansion of the city, the continuous increase in population, and the reduction in forest area make habitat degradation obvious. The patch area of high-grade habitat quality has decreased. Therefore, priority should be given to ecological restoration in areas with degraded habitat quality during protection and restoration. Due to urban expansion, cultivated land and forest land are occupied by cities, and the ecosystem is irreversibly damaged. However, due to social and economic factors, large-scale ecological reconstruction has not met the requirements of the development plan. In order to ensure regional ecological security and ecosystem service supply, targeted ecological reconstruction measures should be taken to adjust the layout of urban green space, set the boundary of prohibited development, and strengthen the construction of urban ecological corridors. In the context of the construction of ecological civilization and sustainable development innovation demonstration area, it is particularly important to coordinate economic development and ecological protection to achieve sustainable and healthy development of Guilin. Deeply understanding the driving factors of habitat quality change will be of great help to promote the construction of the Guilin Sustainable Development Innovation Demonstration Zone and the improvement of habitat quality. Therefore, it is necessary to further explore the driving factors affecting the change in habitat quality in Guilin.

In this study, InVEST model was used to explore the spatial-temporal evolution characteristics of habitat quality in Guilin. Compared with other research methods, InVEST model has been widely used for its simple operation, rapid processing, strong applicability, and other advantages. InVEST model has high accuracy in large-scale habitat quality assessment and has a good spatial effect on habitat quality degradation areas, which can intuitively express the assessment results [16]. However, because its parameter setting depends on the existing research experience and has certain subjectivity, the research results still need to be further verified. In future research, it is necessary to conduct more in-depth research on the driving forces of land cover change and habitat quality change in Guilin, simulate and predict the habitat quality under different scenarios [49,50], which is conducive to revealing the evolution mechanism of habitat quality, improving the habitat quality, and promoting the sustainable and healthy development of Guilin.

## 5. Conclusions

Based on ArcGIS10.5 software (Environmental Systems Research Institute, Redlands, America), Fragstats4.2 software, and InVEST model, this study describes the Spatio-temporal evolution characteristics of habitat quality in Guilin from three aspects of land use change, landscape pattern change, and habitat quality evaluation and further explores the main driving factors of habitat quality change in Guilin by using the method of geographic detector evaluation. The main conclusions are as follows:From 2000 to 2020, the types of land use in Guilin were mainly forest land, accounting for 77.87%, followed by other types of land use, such as cropland for impervious. The changing trend shows that the forest land decreases significantly, the cropland and forest land transform each other significantly, and the impervious area continues to rise, occupying a large amount of cropland, indicating that human activities are the main factor in the transformation of land use.From 2000 to 2020, the irregular shape of patches of various land use types deepened, the fragmentation degree was relatively stable, the landscape diversity was enhanced, and the spatial distribution of patches showed obvious heterogeneity.From 2000 to 2020, the habitat quality of Guilin is mainly of high grade, and the habitat quality is good, but the overall trend is downward. The spatial differentiation of habitat quality in Guilin is obvious, and the northwest is better than the southeast. The habitat degradation area is larger than the upgrading area, and the degradation area is concentrated in the middle and south.From 2000 to 2020, elevation, NDVI, SPLIT separation, and slope are the main factors affecting the habitat quality of Guilin. Because of the undulating terrain of Guilin, forest land is mostly distributed in areas with higher altitudes, and impervious cropland is distributed in areas with lower altitudes; elevation is the main factor affecting the habitat quality of Guilin.


## Figures and Tables

**Figure 1 ijerph-20-00748-f001:**
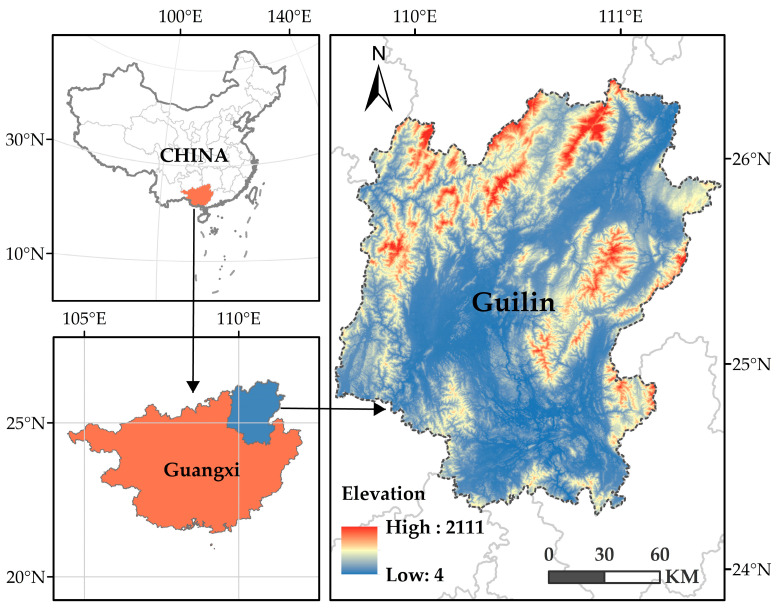
Location and terrain of the study area.

**Figure 2 ijerph-20-00748-f002:**
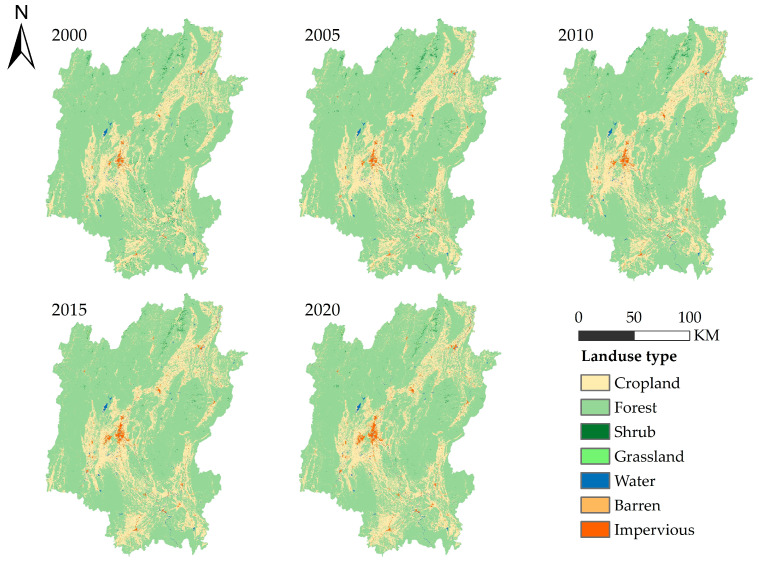
Land use distribution and change in Guilin City from 2000 to 2020.

**Figure 3 ijerph-20-00748-f003:**
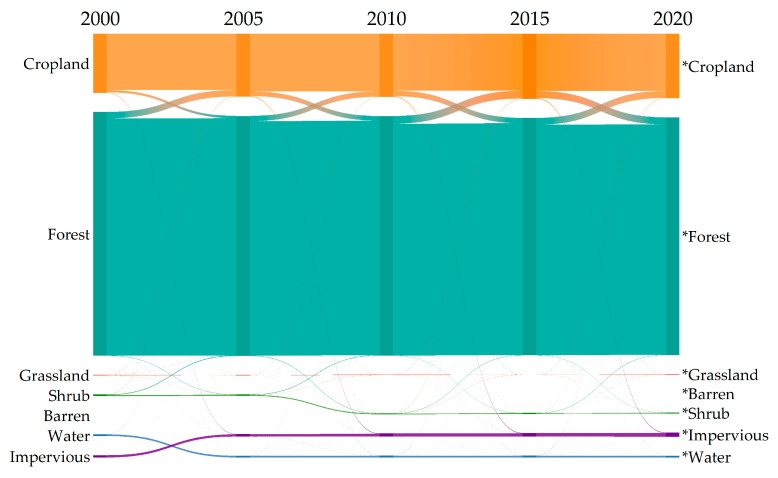
The flowchart of land use change in Guilin City from 2000 to 2020.

**Figure 4 ijerph-20-00748-f004:**
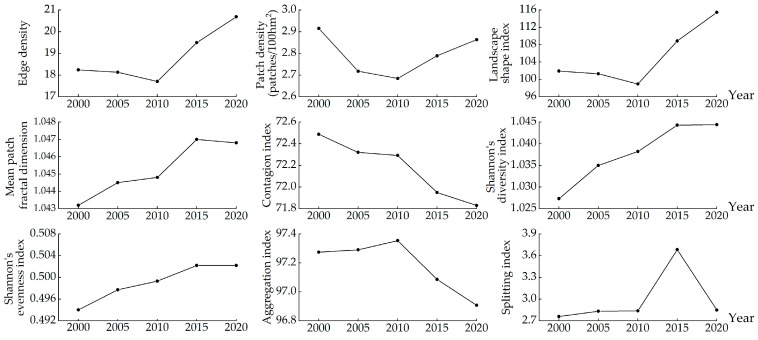
Change in landscape pattern index in Guilin City from 2000 to 2020.

**Figure 5 ijerph-20-00748-f005:**
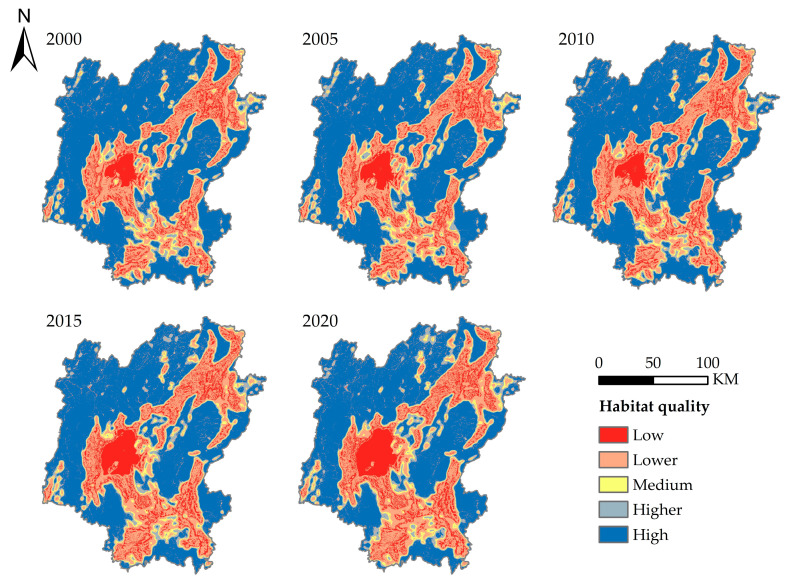
Habitat quality distribution.

**Figure 6 ijerph-20-00748-f006:**
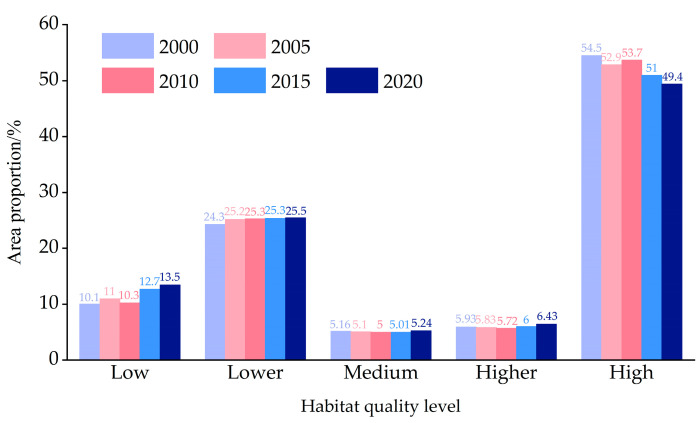
Habitat quality grades (%).

**Figure 7 ijerph-20-00748-f007:**
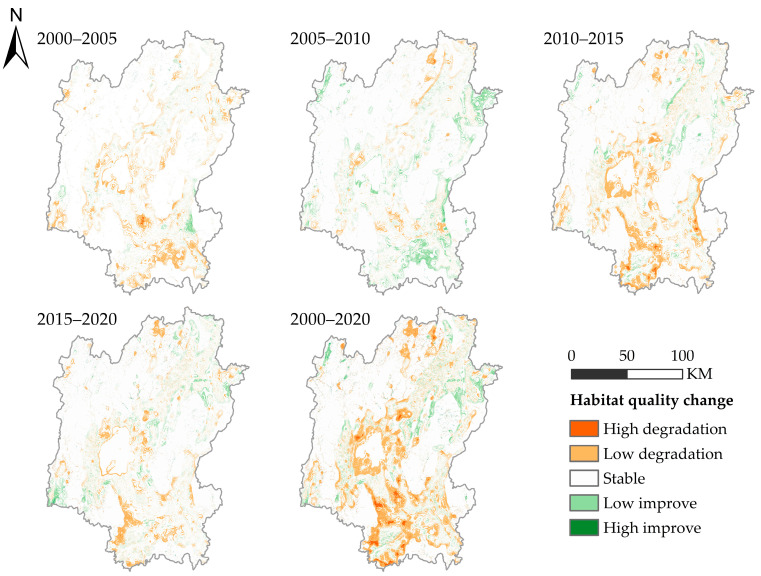
Spatial distribution of habitat quality changes in Guilin City from 2000 to 2020.

**Figure 8 ijerph-20-00748-f008:**
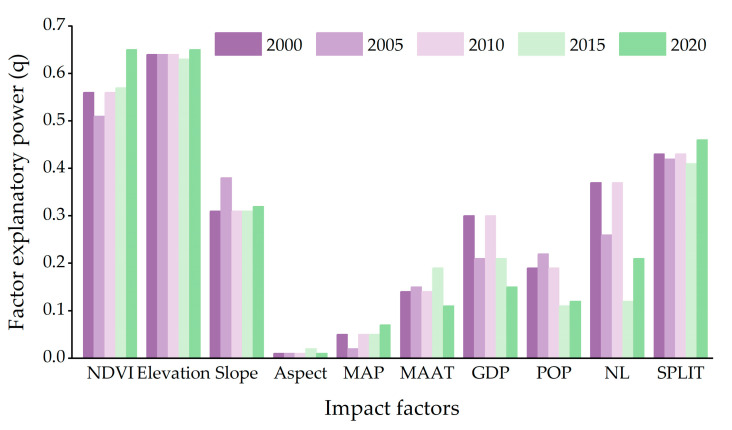
The explanatory power of different influencing factors. (NDVI: Normalized difference vegetation index; MAP: Aspect, mean annual precipitation; MAAT: Mean annual air temperature; GDP: Gross domestic product; POP: Population; NL: Night light; SPLIT: Splitting index).

**Table 1 ijerph-20-00748-t001:** Index selection and its significance of landscape pattern.

Features	Index Name	Significance
Landscape space characteristics	Patch density	It can be used to characterize the patch fineness between landscapes of different sizes. It is the number of patches per unit area
	Edge density	It can represent the degree of landscape boundary separation
	Landscape shape index	It can reflect the shape change in landscape type. The larger the value is, the more complex the shape is
	Mean patch fractal dimension	It can indicate the complexity of the plaque. The higher the value, the more complex the plaque shape is
Landscape structure characteristics	Contagion index data	Represent the degree of aggregation or extension trend of different patch types in the landscape
	Shannon’s diversity index	Characterize landscape heterogeneity and is sensitive to uneven distribution of patch types in the landscape
	Shannon’s evenness index	It can indicate the evenness of plaque
	Aggregation index	It can represent the spatial concentration degree of landscape
	Splitting index	It can indicate the degree of plaque separation

**Table 2 ijerph-20-00748-t002:** Threat factors’ weight and influence distance.

Threat Types	Max Influence Distance	Weight	Decay Rate with Distance
Cropland	3	0.7	linear
Barren	1	0.1	linear
Impervious	6	1	exponential

**Table 3 ijerph-20-00748-t003:** Habitat suitability of different land use types and sensitivity to sensitivity to stress factors.

Land Use and Landcover	Threat		
	Cropland	Barren	Impervious
Cropland	0.4	0.1	0.9
Forest	1	0.6	0.8
Shrub	1	0.5	0.8
Grassland	0.9	0.5	0.7
Water	1	0.5	0.5
Barren	0.1	0.2	0.4
Impervious	0	0	0

**Table 4 ijerph-20-00748-t004:** Transferring matrix of land use types in Guilin City from 2000 to 2020.

		2000							
		Cropland	Forest	Shrub	Grassland	Water	Barren	Impervious	Total
**2020**	Cropland	4329.851	1373.463	23.864	13.245	27.214	0.040	1.431	5769.108
	Forest	777.500	20,354.180	99.257	11.560	3.276	0.000	0.097	21,245.871
	Shrub	2.495	34.212	31.884	1.763	0.000	0.000	0.000	70.354
	Grassland	5.143	8.885	7.069	5.464	0.143	0.005	0.000	26.709
	Water	15.623	3.122	0.008	0.751	121.075	0.000	6.796	147.375
	Barren	0.072	0.042	0.030	0.067	0.000	0.000	0.000	0.212
	Impervious	156.075	20.760	0.106	1.389	4.239	0.005	190.562	373.137
	Total	5286.759	21,794.664	162.218	34.240	155.948	0.051	198.887	27,632.766

**Table 5 ijerph-20-00748-t005:** Changes in land use type area proportion in Guilin City 2000 to 2020 (%).

	2000	2005	2010	2015	2020	2000–2005	2005–2010	2010–2015	2015–2020	2000–2020
Cropland	19.13	20.42	20.46	21.07	20.88	1.29	0.04	0.61	−0.19	1.75
Forest	78.87	77.65	77.48	76.79	76.89	−1.22	−0.17	−0.69	0.1	−1.98
Shrub	0.59	0.47	0.4	0.3	0.25	−0.12	−0.07	−0.1	−0.05	−0.34
Grassland	0.12	0.09	0.09	0.1	0.1	−0.03	0	0.01	0	−0.02
Water	0.56	0.56	0.58	0.57	0.53	0	0.02	−0.01	−0.04	−0.03
Barren	0	0	0	0	0	0	0	0	0	0
Impervious	0.72	0.82	0.98	1.16	1.35	0.1	0.16	0.18	0.19	0.63

**Table 6 ijerph-20-00748-t006:** Landscape pattern index in different years.

Year	Patch Density	Edge Density	Landscape Shape Index	Mean Patch Fractal Dimension	Contagion Index	Splitting Index	Shannon’s Diversity Index	Shannon’s Evenness Index	Aggregation Index
2000	2.9154	18.238	101.9051	1.0432	72.4866	2.7596	1.0273	0.494	97.2742
2005	2.7173	18.1264	101.2875	1.0445	72.3204	2.8309	1.035	0.4977	97.291
2010	2.6845	17.7058	98.9607	1.0448	72.2925	2.8354	1.0382	0.4993	97.3542
2015	2.7893	19.4963	108.866	1.047	71.9487	3.6841	1.0443	0.5022	97.0856
2020	2.8632	20.6893	115.4663	1.0468	71.8293	2.8489	1.0444	0.5022	96.9066

## Data Availability

Data will be available by contacting authors.

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
