# Peer review of "Spatio-Temporal Evolution and Influence Mechanism of Habitat Quality in Guilin City, China"

_ijerph, 2022, doi:10.3390/ijerph20010748_

Round 1

Reviewer 1 Report

This case study has certain reference value, the research logic is reasonable, the research framework is complete. But there are some drawbacks. For example, the pictures in the article are not very detailed and exquisite, and many formats are wrong. In order to better help this paper published. I make a few suggestions.

1. Is there any basis for "ecology and natural resources are recognized as one of the best landscapes in the world" in 2.1?

2. In Figure 1, what is the relationship between the three figures? You need to be clear.

3. In Table 1, what is the basis of classification and why is it judged from these aspects?

4. What is the description and introduction of Figure 3?

5. What is the evidence for this statement in lines 224-229?

6. In Figure 6, the values need to be presented.

 7. In 3.4, the analysis is too little and no effective information can be found.

8. The conclusion is too few, also did not see the valuable information.

Reviewer 2 Report

Thank you for the opportunity to review the manuscript on this interesting and timely topic. The purpose of this paper is to describes the temporal and spatial evolution characteristics of habitat quality in Guilin from three aspects: land use change, landscape pattern change, and habitat quality evaluation, and further explorethe main driving factors of Guilin's habitat quality change. Overall, the structure of the paper respects the “Int. J. Environ. Res. Public Health” format. I can see that the authors gave a significant effort to make the paper well written and compelling. I think this paper can be published once authors make major revisions.

(1). The abstract is not concise enough. Further streamlining is needed to enhance the value of the summary. The author may consider deleting some redundant contents. E.g. lines 27-35

(2). The introduction is not rich enough and must be supplemented. I suggest the author make some necessary additions. The description in this part should directly discuss relevant studies on of Habitat Quality, land use change and of ecological environment. Please consider quoting the following literature as the background of Habitat Quality, land use change and sustainable environment in the introduction.

https://doi.org/10.1007/s11356-021-13444-1

https://doi.org/10.1016/j.uclim.2022.101347

https://doi.org/10.3390/su142416369

https://doi.org/10.1016/j.jenvman.2022.115873

https://doi.org/10.3390/s22239503

(3). In the introduction, lines 69-79 try to introduce the significance of this paper different from other existing studies, but this description does not reflect more innovations. Please consider better sorting and induction.

(4). It seems that the author did not check before submitting. Please carefully check the display errors of 86 line, 120 line, 165 line, 199-207 lines.

(5). There are too many introductions in the method part. Some methods are familiar to readers and reduce their reading interest. Please consider quoting the following literature as the background of method.  For example,2.3.1 InVEST model, please refer to the literature https://doi.org/10.1007/s12145-022-00875-8 , https://doi.org/10.1016/j.ecolind.2022.109632. In 2.3.2. Geographic detector, you should consider quoting https://doi.org/10.3390/land11081303 and https://doi.org/10.1016/j.jenvman.2022.115812. Simply introduce the method.

(6) The pictures of the article are very Nice!

(7) ArcGIS needs to indicate which version it is.

(8). Please consider improving the conclusion and discussion. The discussion part needs to make prospects for this study and the direction of future efforts. I suggest the author delete some unimportant results. A conclusion is like the final chord in a song. It makes the listener/reader feel that the piece is complete and well done. You want them to feel that you supported what you stated in the manuscript. You then become a reliable author for them, and they are impressed by that and will be more likely to read/cite your work in the future.

Round 2

Reviewer 1 Report

I have nothing to say.

Reviewer 2 Report

The author revised the article according to the reviewer's request, and the article has been greatly improved. Articles are suggested. good luck!